# MicroRNA-Based Therapeutic Perspectives in Myotonic Dystrophy

**DOI:** 10.3390/ijms20225600

**Published:** 2019-11-09

**Authors:** Arturo López Castel, Sarah Joann Overby, Rubén Artero

**Affiliations:** 1Translational Genomics Group, Incliva Health Research Institute, Burjassot, 46100 Valencia, Spain; 2Interdisciplinary Research Structure for Biotechnology and Biomedicine (Eri Biotecmed), University of Valencia, Burjassot, 46100 Valencia, Spain

**Keywords:** myotonic dystrophy, microRNA, MBNL proteins, CELF1, miRNA-based drug, miRNA-targeting drug, antisense oligonucleotides, alternative splicing

## Abstract

Myotonic dystrophy involves two types of chronically debilitating rare neuromuscular diseases: type 1 (DM1) and type 2 (DM2). Both share similarities in molecular cause, clinical signs, and symptoms with DM2 patients usually displaying milder phenotypes. It is well documented that key clinical symptoms in DM are associated with a strong mis-regulation of RNA metabolism observed in patient’s cells. This mis-regulation is triggered by two leading DM-linked events: the sequestration of Muscleblind-like proteins (MBNL) and the mis-regulation of the CUGBP RNA-Binding Protein Elav-Like Family Member 1 (CELF1) that cause significant alterations to their important functions in RNA processing. It has been suggested that DM1 may be treatable through endogenous modulation of the expression of MBNL and CELF1 proteins. In this study, we analyzed the recent identification of the involvement of microRNA (miRNA) molecules in DM and focus on the modulation of these miRNAs to therapeutically restore normal MBNL or CELF1 function. We also discuss additional prospective miRNA targets, the use of miRNAs as disease biomarkers, and additional promising miRNA-based and miRNA-targeting drug development strategies. This review provides a unifying overview of the dispersed data on miRNA available in the context of DM.

## 1. Introduction

Myotonic dystrophy type 1 and type 2 (DM1 and DM2) are rare disorders caused by non-coding intragenic repeat tract expansions of CTG (*DMPK* gene) or CCTG (*CNBP1* gene), which are pathogenic above 50 or 75 units, respectively [1]. DM patients have primarily affected skeletal musculature, and display muscle weakness (myopathy), muscle wasting (atrophy), and myotonia as the most recognized signs [2]. DM1 and 2 are characterized as multisystem progressive disorders with the most frequent causes of death being respiratory failure and heart conduction defects. Neuropsychiatric impairment, insulin resistance, gastrointestinal issues, and cataracts are also recurrent clinical features of DM. DM1 patients usually display more severe clinical manifestations than DM2 [2].

It is well accepted that DM is prompted by changes in transcription and messenger RNA (mRNA) processing of multiple genes due to a mechanism involving an RNA-mediated toxic gain-of-function. Specifically, the expanded RNA from *DMPK* or *CNBP1* forms imperfect hairpin structures in DM1 and DM2, respectively [1,3]. These toxic CUG^exp^/CCUG^exp^ RNAs are able to trigger a significant functional reduction of proteins with essential cellular RNA-linked purposes. In DM1, the foremost consequences are the loss-of-function of the muscle blind-like protein family (MBNL1-2) and the gain-of-function of CUGBP Elav-Like Family Member 1 (CELF1) [1,2,3]. On one hand, the CUG^exp^/CCUG^exp^ mRNA accumulates as foci in the nucleus of cells where it binds with high affinity to the Muscleblind-like (MBNL) family of proteins [3,4]. At the post-translational level, there are lower intracellular concentrations and altered splice variant compositions of the MBNL1 and MBNL2 proteins in DM1 muscle precursor cells, which are accompanied by a sustained reduction of MBNL protein during differentiation of myotubes [5]. Thus, the correct execution of MBNL functions is altered in DM, including co-transcriptional RNA splicing and polyadenylation site regulation [6,7]. On the other hand, the levels of CELF1, a protein with roles in RNA processes such as translation, stability, and alternative splicing, are significantly mis-regulated in DM [8,9] through altered mechanisms involving AKT and GSK3β kinases [8,10,11]. Together, the main consequence of MBNL and CELF1 faulty regulation seems to be a failure in fetal-like splicing (and alternative polyadenylation patterns) of hundreds of genes in several tissues and organs, which have been characterized as spliceopathies [3,7]. A functional bond between some of these specific splicing events has been suggested with several key DM clinical phenotypes, such as chloride voltage-gated channel 1 (*CLCN1*) with myotonia [12], insulin receptor (*INSR*) with insulin resistance [13], sodium voltage-gated channel alpha subunit 5 (*SCN5A*), and troponin T2 cardiac type (*TNNT2*) with cardiac conduction defects [14], bridging integrator 1 (*BIN1*), voltage-gated channel subunit alpah1 S (*CACNAS1*), dystrophin (*DMD*), ryanodine receptor 1 (*RYR1*), and pyruvate kinase isozymes M1/M2 (*PKM1/2*) with muscle weakness and wasting [15,16,17], and microtubuleassociated protein tau (*MAPT*) with cognitive decline [18].

Different observations suggest that the critically low levels of MBNL protein activity in DM are a primary contributor to disease symptoms in patients [19,20]. Furthermore, a mouse knockout (KO) for Mbnl1 reproduces myotonia, alterations in alternative splicing, and cataracts [19]. Recently, it has been described that Mbnl1 mouse mutants develop cardiac problems prominent in DM1 (cardiac hypertrophy, interstitial fibrosis, alterations in transcript splicing), which suggests a key role of Mbnl1 in the initiation of cardiac problems in DM1 [20]. However, KO Mbnl1 mice do not recapitulate the full range of DM1 symptoms so it has been hypothesized that Mbnl2 could be compensating for the lack of Mbnl1 function in these mice [21]. When a KO mouse is generated for Mbnl1 in which the Mbnl2 function is also reduced (Mbnl1^−/−^; Mbnl2^+/−^), it was found that these mice are viable but develop cardinal aspects of the disease, including reduced life span, cardiac conduction blockage, severe myotonia, atrophic fibers, and progressive skeletal muscle weakness [21]. A similar central contributing role in the DM phenotype has been suggested for CELF1 [2,3]. Although CELF1 KO mice expressing a toxic RNA transgene did not show correction of DM-linked splicing defects, the normalization of CELF1 prevented in vivo deterioration of muscle function by the toxic RNA, and resulted in improved muscle histopathology [22]. These data suggest that, while reduction of CELF1 may be of limited benefit with regards to DM1-associated spliceopathy, it may be beneficial to the muscular dystrophy associated with RNA toxicity.

The investigation of the pathophysiology of DM has encouraged the development of the first therapeutic strategies such as targeting the toxic RNA (CUG^exp^/CCUG^exp^) structure (hairpin) under the premise of directly or indirectly preventing its formation, or promoting its degradation or blockage by using different types of molecules and technologies [23,24]. Unfortunately, there is still no valid treatment for DM patients. There is only limited success for a few drugs currently in various phases of clinical trials [25,26]. Ideas for innovative therapeutic approaches include boosting the cellular content of MBNL functional molecules, bringing up reduced MBNL activity levels, and the normalization of CELF1 protein levels. Attempts for gene overexpression of muscle blind factors have recently displayed promising therapeutic potential in DM. Specifically, the delivery of recombinant adeno-associated (AAV) vectors carrying MBNL1 or the use of small molecules has been able to recover skeletal and cardiac muscle phenotypes in in vivo, proof of principle studies [27,28,29,30,31]. In parallel, the identification of tideglusib as an effective inhibitor of GSK3β for correcting CELF1 has shown the ability to improve functional aspects in different DM1 mouse models (HSA^LR^ and DMSXL) and patient-derived myoblasts [32]. In this case, expectations are high for the results of the already completed phase II tideglusib DM1 clinical trial (NCT02858908). Together, these results provide solid proof of concept for continuing efforts toward therapeutic gene modulation strategies for DM linked with the correction of altered MBNL or CELF1 expression.

Micro RNAs (miRNA) have lately come to the scene in the pharmaceutical field because of their ability to modulate endogenous gene expression and compensate a given pathogenic dysregulation. Discovered in the mid-1990s, miRNAs are a class of small non-coding RNAs (~22 nucleotides [nt] in length) with robust gene regulatory functions. They promote mRNA messenger degradation, destabilization, or translation blocking (Figure 1A) with assistance from additional protein factors forming the RNA-induced silencing complex (RISC) [33]. A mature miRNA has the ability to bind to the 3′UTRs of mRNAs, even though they occasionally bind to the 5′UTR or even the amino acid coding sequence. Currently, it is estimated that around 1900–2000 miRNAs identified in humans are able to modulate up to 60% of protein-coding genes in our genome [34]. Micro-RNA emergent roles have been linked to key cellular pathways such as stem-cell self-renewal, cellular development, differentiation, proliferation, and apoptosis, etc. [35] with some ubiquitously expressed but others displaying tissue-specific expression. Micro-RNAs have been found to be mis-regulated in numerous diseases, including muscle and neurodegenerative ones [36]. Specifically, miRNA levels can display downregulation or upregulation through a growing number of molecular events (Figure 1B) with already relevant roles identified in cancer, hepatitis C, and diabetes [37,38,39,40,41].

The improved knowledge of miRNA function has attracted tremendous attention in the field of biomedical and pharmaceutical drug development because of their capacity to fine-tune regulatory pathways [40]. Consequently, miRNAs are very suitable targets for manipulating endogenous levels of gene expression to compensate for various disease-specific gene alterations [37,38,39,40,41]. Through the next sections, we describe why miRNAs are stimulating new therapeutic targets in DM with special attention paid to their connection with MBNLs and CELF1 key disease factors. In addition, we present a description of the current picture of miRNA dysregulation in DM.

## 2. Micro-RNAs and Myotonic Dystrophy

As mentioned above, miRNA dysregulation is growing in relevance in various human diseases including systems and tissues severely affected in DM. Micro-RNA evaluation was recently addressed in DM in order to decipher whether they are contributing to its complex pathogenesis. In this section, we summarize the first studies performed and results already published, introducing promising pieces of data for developing miRNA-based therapeutic approaches, and highlighting current miRNA dysregulation in DM (Table 1).

### 2.1. Therapeutic Intervention of miRNAs in DM

Drosophila melanogaster provided the first evidence for the possibility of substantial correction of functional DM disease phenotypes based on miRNA-based established technologies (see technology description in Figure 1C). By manipulation of a DM1 fly model expressing 480 CTGs, Fernández-Costa et al. (2013) first proved that over-expression in the musculature of dme-miR-10, previously found downregulated in DM1, was able to partially recover the reduced lifespan characteristic of the model flies. However, the intrinsic mechanism was not elucidated [52]. Later, Cerro-Herreros et al. (2016) used miRNA sponge constructs to block dme-miR-277 and dme-miR-304, previously identified as 3′UTR translation repressors of mbl mRNA in the fly, to increase endogenous muscle blind levels (Figure 2). A muscle blind increase was demonstrated not only in a wild type background but also in the DM1 fly model expressing non-coding CUG trinucleotide repeats throughout the musculature. This approach was sufficient to rescue DM1 mis-splicing events as well as lessen muscle atrophy. Importantly, the positive molecular and histological results triggered a functional improvement in deficient climbing and flight ability, and a significant increase in their shorter lifespan. Together, Drosophila provided an interesting proof-of-principle for the therapeutic upregulation of muscle blind by targeting defined miRNAs in humans [58]. Unfortunately, it was not possible to check if the same MBNL modulation could be attempted in DM1 patient-derived cells since both miRNAs are not conserved in humans.

The first attempts at modulation of conserved human miRNAs are also available in the context of DM (Figure 2). The limited available data is mainly connected to MBNL and CELF1 cellular roles. Rau et al. (2011) [51] reported that, in heart tissue from DM1 patients, MBNL1 protein sequestration results in the further decreased expression of mature miR-1 levels and increased amounts of its gene targets. This paradigmatic model connects the biogenesis of a specific miRNA with a central pathogenic event in DM1 cells. Specifically, MBNL1 was found to bind to the conserved UGC motif in the pre-miR-1 loop and to compete with the LIN28 protein for binding in the same site. Because LIN28 is a Dicer competitor, sequestration of MBNL1 allows extra LIN28 to bind to the pre-miR-1, triggering its uridylation and inefficient processing by Dicer [51]. In the same study, decreased expression of miR-1 is linked to specific DM1 symptoms through the upregulation of direct targets of this miRNA. One of the miR-1 targets, the CACNA1C protein, which was previously identified as responsible for arrhythmias and sudden death when mutated [67], was found significantly upregulated in the heart of DM1 and DM2 patients [51]. Lastly, in order to demonstrate the potential use of miR-1 as a therapeutic target, increasing miR-1 levels in normal H9C2 cardio-myoblasts by co-transfection with a miR-1 mimic was able to modulate (reduce) the endogenous levels of the CACNA1C factor [51]. MiR-1 was also found to be downregulated in a Drosophila DM model after mbl was silenced [52].

On a similar theme, a different miRNA-MBNL connection has also been established. Zhang et al., (2016) identified the miR-30-5p family as direct repressors/inhibitors of MBNL1-3 translation [66]. In DM1, the alteration of MBNL1 functional levels triggers the mis-regulation of the alternative splicing of Trim55 and Insr muscle-related genes. The Trim55 gene encodes a muscle-specific protein involved in sarcomere assembly [68]. The binding of insulin to INSR initiates the PI3K/AKT (protein kinase B) and ERK (a kinase of the MAPK family) muscle signaling pathways. These are connected to the DM scenario via the MEF2 transcription factor that directly regulates muscle development [66,69,70,71]. In this case, the researchers proved that the introduction of miR-30-5p (mimic product) in standard muscle C2C12 cells was able to have the anticipated effect (dysregulation) on the alternative splicing of Trim55 (exon9+) and INSR (exon11+), and was able to downregulate the expression of MBNL reporters. This proved a functional connection between the miRNA and DM. These results open the door to evaluate the therapeutic ability of miR-30-5p inhibition by the use of antimiRs to increase MBNL protein levels and restore DM-linked splicing alterations.

Cerro-Herreros et al. (2018) also recently identified miR-23b and miR-218 as regulators of MBNL1 and MBNL2 factors after a screening in HeLa cells. Specifically, these two miRNAs are repressors of MBNL translation. After miRNA blocking with antagomiR treatments for each miRNA independently, it was possible to enhance MBNL protein levels, restore their normal distribution in the cytoplasm and the nucleus, and rescue pathogenic mis-splicing events in DM1 myoblasts [61]. Both miRNAs were also found to be highly expressed in tissues relevant to DM1 in mice (skeletal muscle, heart, and CNS). To further investigate their role, miR-23b and miR-218 were investigated for their in vivo therapeutic potential by designing and delivering miR-23b and miR-218 antagomiRs, with very promising results. In a DM1 disease mouse model (HSA^LR^), which lacks the *DMPK* transgene and specifically overexpresses 250 untranslated CUG repeats in the skeletal muscle after systemic delivery (three subcutaneous injections to a final dose of 12.5 mg kg^−1^) of synthetic antagomiRs against either miR-23b or miR-218, Mbnl1 and Mbnl2 expression in skeletal muscle went up by approximately two-fold and four-fold, respectively. These results were linked with a pronounced in vivo response in disease-linked phenotypes such as the significant improvement of the aberrant spliceopathy profile, histopathology disease signs, and functional myotonia without any clear toxicity issues. Importantly, MBNL1 and 2 protein over-expressions were well-tolerated as mice showed no detrimental phenotype six weeks after the treatment [61]. This miRNA targeting approach displayed similar MBNL1 protein increases in the same HSA^LR^ mouse model in comparison with the previously described strategy of the adeno-associated virus (AAV) vector transduction for MBNL1 over-expression [27]. However, the miRNA approach has a particular advantage. By invoking simultaneous MBNL1 and 2 modulation, the miRNA approach can achieve significant and long-term improvement of DM1-linked phenotypes, including not only mis-splicing and myotonia, but also histopathology (central nuclei) and muscle strength recoveries. In addition, small changes in miRNA expression were capable of large and significant phenotypic effects. Due to this, it may be possible to avoid regulatory issues associated with AAV-linked drug developments and achieve a tissue-specific gene modulation through the fine-tune control of regulatory miRNAs’ intensity.

MiRNA-based therapeutic strategies are also being investigated for the CELF1 factor. In this case, miR-206 has focused mainly on DM skeletal muscle. First, Koutalianos et al. (2015) [49] described the alteration of myogenesis in DM1 cells connected to the following mechanism: miR-206 downregulation, upregulation of the Twist-1 gene target, and final MyoD downregulation. The same message is obtained from recent Dong et al. (2019) work [49], adding that the alteration of the miR-206 levels also have the ability to modulate CELF1 overexpression. In the older study, researchers externally induced downregulation of miR-206 using an antagomiR molecule, which was validated by the observed increased levels of TWIST-1 protein, which is a direct gene target, in normal human myoblasts. Furthermore, in three different cell lines modeling DM1, they recovered the low miR-206 levels by transfection with a miR-206 mimic, which was able to recover deficient myotube formation by quantifying the cell fusion index mediated by the achieved upregulation of its Twist-1 target [49]. The same miR-206 mimic replacement strategy, in cells overexpressing Celf1, significantly improved myoblasts fusion index and the myotube area by inhibiting Celf1 expression [49]. This result is a new therapeutic opportunity to be added to the developing field with miRNA intervention of altered CELF1 overexpression in DM1 [8,11].

Concerning the altered cardiac function in DM, Kalsotra et al. (2010) [49] indicated that Celf1 post-transcriptional repression during development was mediated by the upregulation of miR-23a and miR-23b in normal cardiomyocytes. In a different study, they wanted to determine whether miRNA altered expression in DM1 could be an additional source of CELF1 upregulation [48]. Using an established heart-specific DM1 mouse model (EpA960), which expressed human DMPK exon 15 containing 960 CUG repeats, they found that miR-23a and miR-23b levels were dramatically down-regulated in the hearts of adult mice after inducing the toxic CUG^exp^ RNA, which led to increased CELF1 levels [48]. For the latter study, miR-23 replacement in a DM background was not attempted to evaluate potential therapeutic benefits. Kalsotra el al. (2014) analyzed >500 additional miRNAs to identify miRNA alterations caused by the toxic repeat expansion. Several other miRNAs were found to be downregulated, including miR-1, miR-133, miR-499, miR-29, miR-30, and miR-133, as well as upregulated (miR-21). Low expression was conserved for these miRNAs on a set of 22 human DM1 hearts [48], where some were previously associated with arrhythmias or fibrosis [72,73,74]. Nevertheless, the miRNA expression profile induced by CUG^exp^ was not reproduced by loss of Mbnl1 or gain of Celf1 alone. However, functionally, they demonstrated strong downregulation of MEF2 in adult DM1 hearts and many of the altered miRNAs are direct transcriptional targets of MEF2. Mis-regulation of miRNA expression was rescued by the introduction of new MEF2C protein displayed as an additional means to target miRNA modulation in DM. However, the specific mechanism by which CUG^exp^ RNA affects mRNA and protein levels of MEF2 paralogs was not shown.

Lastly, Capella et al. (2018) performed an innovative sequencing screen of RISC-associated mRNAs for identifying those displaying expression dysregulation between primary skeletal muscle fibroblasts derived from DM1 patient´s biopsies and matched controls [53]. In the second step, a bioinformatics analysis defined the miRNA/mRNA relevant interactions. From 24 correlations found, the study focused on miR-29c and its ASB2 target. Low expression of the miRNA and elevated ASB2 target levels were then further confirmed in DM1 cell lines and biopsies and established as a functionally relevant miRNA/mRNA interaction for DM1 muscle pathology. Similarly, miR-29 was also formerly reported as significantly changed in DM1 hearts [48], blood, and skeletal muscles in DM1 patients [43,50]. The ASB2 target was also reported to be involved in regulating muscle mass [75]. MiR-29 deregulation in skeletal muscle samples has an impact on the expression of additional predicted targets such as TRIM63/MURF1, DIABLO, RET, and TGFB3, which relate the ubiquitin-proteasome system and apoptosis to atrophic muscles [76,77,78]. These results point to the involvement of miR-29 reduction in DM1 myofiber atrophy [43]. A therapeutic opportunity was shown by using CRISPR/Cas9-mediated deletion of CTG expansions from DM1 myogenic cell clones that ultimately recovered normal miR-29c and ASB2 levels, which indicates an additional direct link between the mutant repeats and the miRNA/target expression, as established in DM1 hearts [48].

### 2.2. Picture of miRNAs Dysregulation in DM: Potential as a Disease Biomarker

In addition to therapeutic use, miRNA characteristics make them good candidates for diagnosis/prognosis biomarkers for diseases. These features include molecular stability in-body, easy detection in both blood and plasma, and stability after external manipulations [79,80]. The pharmaceutical field has achieved many advances in the miRNA-based diagnostic area, with products already in the market or in advanced (clinical) evaluation. However, there are still no miRNA-based or miRNA-targeting therapeutic drugs in the market [40].

In DM, several studies have focused on proving that miRNAs are dysregulated in samples such as skeletal muscle biopsies, blood, serum, or plasma from DM1 patients (mainly adult form) when compared to the same type of samples from control individuals [42,43,44,45,46,47,50,54,81,82]. A rational starting point has been performed over muscle-specific myomiRs (miR-1, miR-133a, miR-133b, miR-206, miR-208b, miR-486, and miR-499). MyomiRs, which account for approximately 25% of all miRNA expression in the muscle, are a selection of miRNAs highly enriched in muscle. They have tight control over all aspects of muscle homeostasis as well as in myofiber control type, muscle hypertrophy, and atrophy processes and have previously shown differences in expression in other muscle diseases [83,84,85]. Additional myogenesis mechanisms have also been shown to be regulated by myomiRs including splicing during embryonic myogenesis, where miR-133 directly downregulates the key splicing factor nPTB, which is the neuronal homolog for polypyrimidine tract-binding protein [86]. The myomiR results showed that miR-206, miR-1, and miR-133a/b usually display disrupted levels (downregulation) and/or cellular mis-localization in DM1 biopsies [42,43,44,50]. Likewise, in the first attempts to identify valid circulating DM1 biomarkers, the same myomiRs were analyzed and found significantly upregulated in blood, serum, or plasma from DM1 patients [45,46,47,54]. In all cases, they efficiently discriminated DM1 patients from controls and, in some of the studies, the data offered good correlations with patients’ muscle strength levels (controls: healthy individuals) or ongoing waste (controls: stable DM1 patients). Together, the anomalous high levels of these miRNAs, also reported in other rare muscular dystrophies as Duchenne´s disease (DMD), seem to correspond with low muscle strength function [87].

In spite of the above reliable connection, the data still reveals some inconsistencies. The most striking is the negative finding of miRNA level differences between DM1 and controls in other studies [81,82]. These results reveal the challenges that still need to be solved concerning miRNA methodological study designs similar to the cancer field [56]. Together, methodologies suffer from low sample size, different type of samples analyzed, or the use of dissimilar approaches for miRNA level normalization, among others. A paradigmatic example is the generalized notion that miR-16 levels are stable. However, miR-16 has been recently observed to be sequestered by long CUG repeats [64], which precludes its use in the DM1 field as a valid miRNA for normalization purposes. Consequently, the amount of free miR-16 in the bloodstream may not be equal in healthy and DM1-affected individuals. Suggested solutions for resolving miRNA quantification divergences among studies include increased sample size, consensus on the normalization approaches, identification of the mechanism leading to changes in a miRNA level, supporting conclusions on animal models, and additional methodologies such as digital droplet PCR (for absolute number quantification) or ribo-profiling. One example is how the simultaneous use of different normalization approaches have further solidified candidates for the same type of DM samples [45,54]. For example, a contradiction was found in two independent studies for miR-1 mis-regulation in DM1 biopsies. The contradiction was due to it being upregulated in Reference [43] but downregulated in Reference [42]. Supporting data achieved from additional biopsies [52] and from the conserved miRNA in Drosophila, in a model where 480 CTG repetitions are expressed [52], established a consensus for miR-1 downregulation in DM skeletal muscle.

Although fewer in number, similar myomiR studies have also been approached in DM2. These studies did not show myomiR dysregulation [43,55], but similar trends to DM1 studies were observed [43]. Mis-matching results could be due to the low number of patients examined in those studies. Other speculative reasons could be due to the difference in the genetic mutations causing DM1 and DM2 or possibly the relatively milder DM2 phenotype. Likewise, to complete the whole miRNA dysregulation picture available in DM, miRNAs other than myomiRs were also found to be mis-regulated over biopsies and blood patient´s samples in DM1 and DM2 backgrounds. In this case, the level of confidence for specific miRNA dysregulation is lower than for myomiRs, as most of the data come from single studies (Table 1). Only a few myomiRs, previously involved in disease, such as miR-29 and miR-33, are validated from independent studies [43,50,53]. Again, it was unexpected that the list of miRNA candidate biomarkers for DM2 was almost completely different from the ones identified from DM1 studies even though they are very similar disorders. There were only a few overlapping matches (miR-208a, miR-193b, miR-1, and miR-381), which came from two different studies [51,55].

## 3. Future Prospects for miRNA-Based Therapeutics in DM

The successful achievement of a valid miRNA-based therapeutic approach for DM patients, still in very premature phases, will need further research and progression from two different avenues. One is in connection with typical challenges faced in every drug development process. The second will require new and innovative approaches to combat the disease. In this case, we define them in more depth.

### 3.1. MiRNA-Based and miRNA-Targeting Drug Development Challenges

Despite its potential, development of miRNA-based and miRNA-targeting drugs still needs time and technical breakthroughs, especially with regard to stability and delivery issues that complicate the development of miRNA therapeutics and arrival of validated drugs in the market. For this reason, most of the technologies are still in preclinical phases, with only a few molecules undergoing clinical evaluation in fields like cancer or hepatitis with a strong knowledge about the disease [40]. Regarding DM, miRNA-based and miRNA-targeting drug development is even farther behind when compared with these other diseases. This situation may be linked to DM´s rare disease status, where the development of therapeutics was neglected for a long time, as well as other issues such as the high level of disease complexity, with no clear endpoints for drug evaluation. Another issue is the recent understanding of the genetic causes of the disease as well as the identification of a highly novel pathogenic mechanism with potential disease targets linked to sequestration by a toxic RNA. However, the first proof-of-concept results described in the previous sections are the real starting point for achieving a safe and effective treatment for DM patients. Even so, the final use of miRNA-based drugs will pass through the challenge of identifying the most efficacious therapeutic candidates and the evaluation of new antisense oligonucleotide technologies with different chemistry and delivery options.

Currently, examples of the most developed approaches in other disease fields include antagomiRs [88] and miRNA mimics [89], for which there is also proof-of-concept in DM [49,51,61,62,63,64,65,66]. However, the use of blockmiRs (or miRNA masking approach) (Figure 1) may be a very interesting alternative therapeutic option for target upregulation. Since each miRNA is able to regulate hundreds of genes, the action of antagomiRs is recognized as “sequence-specific,” which is a feature that causes off-target side effects and unwanted toxicity. However, a blockmiR is a “gene-specific” option to achieve specific mRNA target upregulation with exquisite specificity and low undesirable off-target effects. One promising example is in connection with myomiRs miR-1 and miR-133, which are found altered in DM. BlockmiRs complementary to cardiac pacemaker channel encoding genes, HCN2, and HCN4, where miR-1 and miR-133 bind, prevented the repressive actions of both miRNAs on protein expression of these genes and caused a positive acceleration of the heart rate in a rat model [90].

The initial in vivo studies with miRNA-targeting products frequently resulted in little success, which displays weak therapeutic outcomes [37]. These studies require repeated administration to achieve persistent miRNA inhibition, have high production costs, and/or show low efficiency in some tissues and cell types. These effects may be a consequence of high levels of molecule degradation in the bloodstream and/or poor delivery to the final target site [37]. Therefore, the correct delivery of the oligonucleotide product to the targeted organs in order to maintain adequate treatment specificity can require passive or active strategies. Targeting of organs like the liver, spleen, and lymph nodes take advantage of their tendency to internalize accumulated particles (nanoparticles or liposome-like particles) that incorporate the oligonucleotide molecule. However, different organs or systems, such as skeletal muscle heavily affected in DM, will need the use of specific binding molecules to activate the endocytosis in the cells of interest. One of the major issues for all RNA-based therapeutics is that these molecules are very unstable and prone to degradation by RNases because of their 2′-OH chemical group. Recent strategies for increasing stability of antimiRs, as well as improving cell intake and tissue distribution, have introduced different types of chemical modifications previously developed for ASOs, such as 2′-O-methoxyethyl (2′-MOE), 2′-fluoro, phosphorothioate (PS), or locked nucleic acid (LNA), among others. Modified antimiRs currently under evaluation display, in some cases, improved target modulation compared with unmodified antimiRs with promising results in in vivo models of cancer, cardiac disease, and diabetes, and in non-human primates [37]. Similarly, different chemical modifications could provide interesting improvement opportunities for the already promising miRNA-based and miRNA-targeting anti-DM molecules identified for the therapeutic intervention of miR-1, miR-206, miR-23b, or miR-218 levels [49,51,61]. Otherwise, beyond the practice of basic in vitro transfection methodology options for the development of “miRNA-gain-of function,” in vivo treatments include the use of viral transduction of pri-, pre-, or mature miRNAs. Different studies have demonstrated the in vivo validity of this strategy by reintroducing specific miRNAs that achieve a fine-tune miRNA product expression to block lung and liver cancer processes [91,92]. Depending on the viral vector type and the proliferation status of the target cells, DNA-encoded approaches may be continuously expressed, facilitating a prolonged miRNA increase, as well as a suppressive response [91,92,93]. However, safety issues still need to be resolved for this technology [93]. This last limitation has led to the development of encapsulating nanoparticle approaches following the knowledge gained from the development of siRNA delivery methods (very similar in structure and functions) to increase the efficacy of in vivo delivery of miRNA-based drugs [37]. Thus, the use of poly(lactide-co-glycolide) (PLGA), TargomiRs, N-acetyl-D-galactosamine (GalNac), or synthetic polyethyleneimine polymer particles, some already in clinical trial evaluation for cancer or diabetes disease, is also suggested for anti-DM miRNA-based and miRNA-targeting drug developments.

### 3.2. DM Drug Development Opportunities for miRNA-Based and miRNA-Targeting Products

The different attempts for developing a treatment in DM have shown how intricate it can be due to the high complexity of the disease. This is why the incorporation of additional and innovative strategies is desired. Only recently have proof-of-concept miRNA-based therapeutic strategies been tested, like blocking endogenous miRNAs [61] or delivering exogenous miRNAs by mimetic or miRNA-encoding expression vectors [49,51]. However, they were usually presented as short experiments in the context of larger basic DM-linked research approaches. Tangible advances in rational drug developments are a prerequisite for further therapeutic success, which is already ongoing in diseases like cancer or diabetes with miRNA-based and miRNA–targeting products already in clinical evaluation. Based on this, the consistent identification of myomiRs as mis-regulated in DM [42,43,44,45,46,47,50,54] is offering an exciting starting point for miRNA intervention approaches, be they individually modulated or in combination. After muscle injury, the treatment of mice with a combination of three myomiRs, but also with other miRNAs, led to enhanced muscle regeneration and prevented fibrosis of regenerating muscle [94].

Basic but innovative DM research strategies involving miRNAs are promising. One example is the transfection with one artificial mirtron. Mirtrons are introns that form pre-miRNA hairpins after splicing to produce RNA interference effectors distinct from Drosha-dependent intronic miRNAs. After transfection, the mirtron causes a successful functional knockdown of the mutated DMPK gene in a murine myoblast line containing a pathogenic copy of the gene with more than 500 CUG repeats. Correction of Serca-1 mRNA DM1-associated splicing abnormality was detected, which demonstrated the therapeutic potential of this RNAi strategy [95]. In addition, linked to the use of miRNAs as a strategy for targeting the initial DM mutation is that Dmpk expression might be directly regulated by different miRNAs, such as miR-1, miR-206, and miR-148a [64]. Importantly, their results include a possible cooperation between miRNAs and discuss gene targeting by miRNA pairs that vary in distance between their binding sites and expression profiles. Roles for miR-15b, miR-16, and miR-214 are also described when proposing that the CUG^exp^ may serve as a target for concerted regulation by miRNAs and may also act as molecular sponges for natural miRNAs with CAG repeats in their seed regions, which affects their physiological functions [64].

Until recently, the use of small molecules for the direct targeting and modulation of specific miRNAs was not possible. However, this avenue has been opened by identifying azobenzene as a specific and efficient inhibitor of the biogenesis of miR-21 [96]. From there, several other small molecules have been identified as able to modulate miR-21 and miR-31 [97,98]. These molecules have mechanisms of action linked to the direct binding of the pre-mir-21 hairpin and interference with further Dicer processing. One interesting subject would be to study if some of the several small molecules in development are able to modulate miRNA profiles implicated in DM. This could identify unique tools for investigating miRNA roles in DM as well as promising reagents to further boost patient response to ongoing treatment developments.

In addition, given that DM1 and DM2 are multisystemic disorders, investigating tissues other than skeletal muscles or heart would also be interesting to elucidate miRNAs’ contributions to other DM clinical symptoms. Some pieces of data have recently become available from DM-linked cancer and cataracts. Fernández-Torrón et al. (2016) described that miR-200/141 downregulation was linked to higher cancer risk observed in DM1 patients as the first route of explanation for this unexpected feature [57]. Likewise, a study regarding cataracts, frequently found in DM patients, was published presenting a prediction of miRNA networks in DM1 and DM2 for the genes differentially expressed in lens epithelial samples from patients [98]. Notably, only the predicted miRNAs miR-197 and miR-29c were shared between both pathologies at this level. Furthermore, miR-29 and miR-133, predicted in the eye in this study, were previously found as mis-regulated in other DM tissues [43,44,45,48,49,50,53,54], which makes these miRNAs very promising for overall disease observation and intervention. At this point, no studies have been published involving the potential connection between miRNAs and CNS disabilities described in DM patients. In addition, being that the majority of (therapeutic) studies in DM currently focus on type 1, it would be advantageous to give more attention to type 2. Because they are so similar in nature, some of the miRNA-based approaches could be of common benefit to both types of DM. This rationale could also be applied to other rare diseases with the same mutation starting point (a CTG expansion) and/or with similar MBNL loss of function such as in spinocerebellar ataxia type 8 (SCA8) [99] and Fuch’s dystrophy [100]. There is recent evidence that miRNA dysregulation can potentially contribute to disease pathogenesis in other repeat expansions disorders (in addition to CTG sequence) like Huntington’s disease (HD), Fragile-X syndrome (FXS), and spinocerebellar ataxias (SCAs) (reviewed in [101]). These diseases also lack conventional therapies. It is still too early to tell if miRNA dysregulation is common for repeat expansion diseases. Therefore, a better understanding of miRNA function and biogenesis may lead, as expected in DM, to the development of new therapeutic strategies for preventing or delaying the disease.

Since MBNL1 and 2 and CELF1 are the first line of therapeutic targets in DM disease through fine-tuning regulatory miRNA expression levels, renewed efforts are needed to identify additional miRNA candidates and further develop those already identified. Looking at the literature, it was recently published that the miR-322/-503 cluster has been linked to cardiomyocyte specification and an early cardiac fate, likely by targeting Celf-1 [63]. This data suggests that these miRNAs play a potential role in DM1 and would be an interesting route to explore.

## 4. Conclusions

This overview of the recent connection between miRNA and myotonic dystrophy offers two important messages. First, there is a more than probable pathological role of miRNA dysregulation in muscular issues displayed by DM patients. Second, because no conventional therapy exists for DM, a better understanding of the miRNA machinery and miRNA functions will help develop new treatment strategies for preventing or delaying the muscular degenerative process underlying DM. Proof of concept miRNA-based therapeutic formulations have shown promise, with in vivo approaches exhibiting a low toxicity profile and successful delivery to the muscle site. As the demand for a valid treatment grows, future prospects in the field must connect the current surge in genomic and proteomic data in human biology with the identification of key solid miRNA targets for drug development. The confirmation of these targets, coupled with the comprehensive use of novel delivery platforms and improved molecules, should enable miRNA therapeutics to become a viable clinical reality for DM patients.

## Figures and Tables

**Figure 1 ijms-20-05600-f001:**
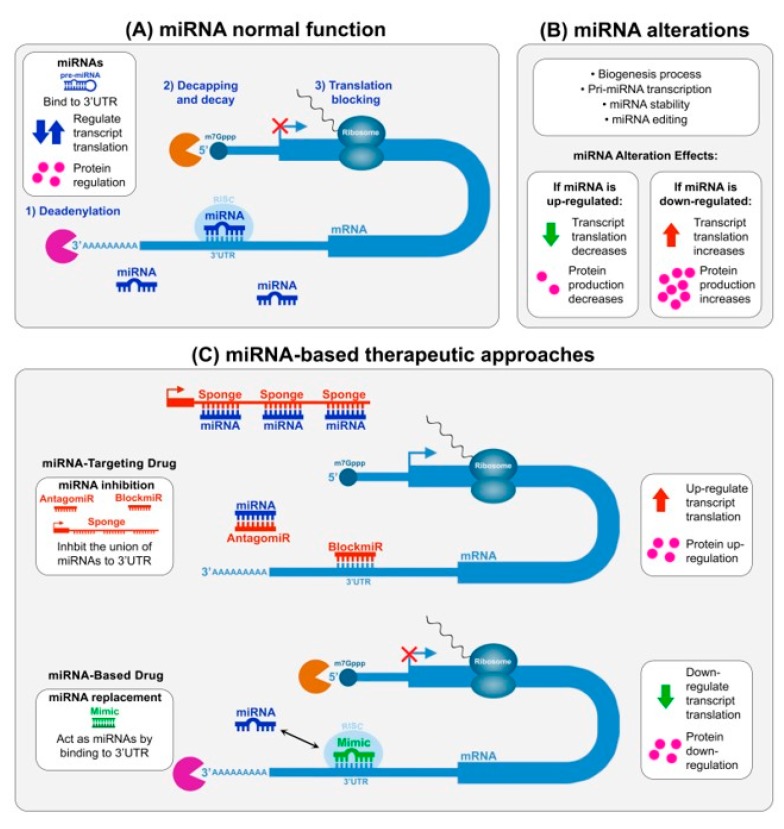
Micro RNA (miRNA) mechanisms for mRNA translation, regulation, and therapeutic intervention. (**A**) miRNA normal function. In mammals, the interaction between miRNA (in blue) and mRNA targets trigger different mechanisms for transcript regulation through the RISC complex in order to achieve normal cellular protein levels (pink circles) [36,37]. (**B**) Causes of miRNA dysregulation. Alterations in miRNA biogenesis, editing, or in its biological stability may cause pathological upregulation or downregulation, which leads to decreased (green arrow) or increased (red arrow) target transcript translation regulation and final protein levels (pink circles), respectively [36]. (**C**) Illustration of miRNA-based technologies [36,37,59]. There are two main strategies of miRNA intervention depending on what is needed with regard to miRNA level correction. (Upper panel in **C**) When a miRNA is upregulated, inhibition is conducted by using antimiR products (in red) after miRNA-targeting drug development. Different types of antimiR products exist based on their mechanism of action. AntagomiR synthetic molecules are antisense oligonucleotides (ASOs) perfectly complementary to the specific miRNA target. A second strategy is the use of blockmiRs, which are designed to have a sequence that is complementary to one of the mRNA sequences that serve as a binding site for a microRNA. Upon binding, blockmiRs sterically block the microRNA from binding to the same site, which prevents degradation or transcription inhibition of the target. A third approach for direct miRNA binding and enhanced levels of inhibition involves the use of miRNA sponges. Sponges contain several tandemly arranged miRNA target sequences (same or different ones) usually embedded in the 3’UTR of a reporter gene for assessing the activity [60]. (Lower panel in **C**) miRNA replacement is conducted to restore its function by introducing a miRNA mimic product (in green) and, thus, following miRNA-based drug development. Micro-RNA mimics are synthetic double-stranded biomolecules that contain one strand with the same sequence and chemistry of the lacking miRNA and a second complementary strand that contains chemical modifications used for the delivery and protection of the mimic.

**Figure 2 ijms-20-05600-f002:**
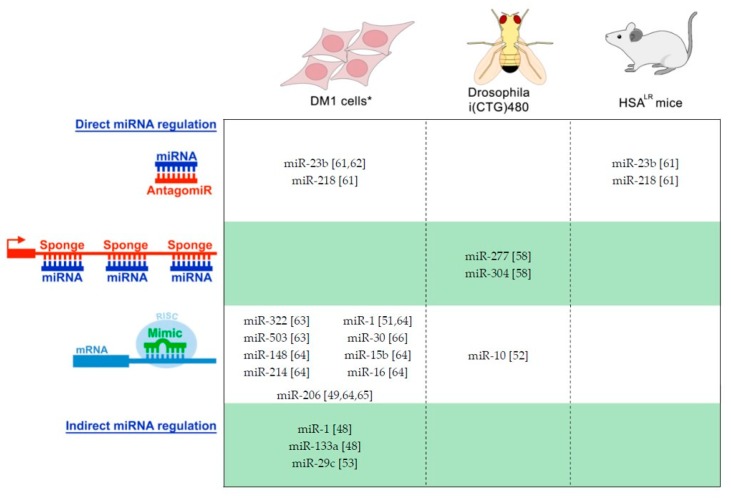
Therapeutic proof-of-concept approaches for DM based on the modulation of miRNA levels. Currently, three different model systems have been used for evaluation: cells* (human and murine lines), flies, and mice. In vivo miRNA interventions were performed in disease backgrounds to directly assess the therapeutic potential in DM. * Cell approaches used DM and non-disease lines indistinctly for therapeutic evaluation and for conceptual modulation of DM-related targets, respectively. As direct miRNA intervention technologies, antagomiRs, sponges, and miRNA mimic products, able to directly bind to miRNA targets, have been used [49,51,61,62,63,64,65,66]. But miRNA modulation has been achieved by indirect approaches such as the recovering of MEF2C levels in DM1 [48] or removal of the expanded CTG by CRISPR technology [53].

**Table 1 ijms-20-05600-t001:** Altered miRNAs in DM1 or DM2.

miRNA	Alteration	Mechanism	Target	Disease Role Suggested	Refs.
***DM1***					
miR-206	↑ (sk)			Muscle atrophy	[42]
	≠cd (sk)				[43]
	↑ (sk)				[44]
	↑ (pl)			Muscle strength	[45]
	↑ (se)			Progressive wasting	[46,47]
	↓ (*m*h, h)	MEF2		Arrhythmias/Fibrosis	[48]
	↓ (sk)	MYOD	↑ TWIST-1	Muscle differences	[49]
miR-1	↑/≠ cd(sk)			Muscle development	[43]
	↓ (sk)		↑ several transcripts		[44]
	↓ (sk)				[50]
	↓ (*m*h, h)	MBNL1/LIN28	↑ GJA1a/↑CACNA1C	Cardiac dysfunction	[51]
	↑ (pl)			Muscle strength	[45]
	↑ (se)			Progressive wasting	[46,47]
	↓ (*m*h, h)	MEF2		Arrhythmias/Fibrosis	[48]
	↓ (*d*m, sk)	Mbl			[52]
miR-335	↑ (sk)				[43]
miR-29b,c	↓ (sk)		↑ several transcripts	Atrophy	[43]
mirR-29c	↓ (sk)				[50]
mirR-29b	↑ (bl)				[50]
miR-29c	↓ (sk)		↑ ASB2 (and others)	Muscle fibrosis/mass	[53]
miR-33	↓ (sk)				[43]
mir-33a	↑ (bl)				[50]
miR-133b	≠ cd (sk)				[43]
miR-133a/b	↓ (sk)				[44]
miR-133a	↓ (sk)/↑ (bl)				[50]
miR-133a	↑ (pl)				[54]
miR-133a/b	↑ (pl)			Muscle strength	[45]
miR-133a/b	↑ (se)			Progressive wasting	[46,47]
miR-133a	↓ (*m*h, h)	MEF2		Arrhythmias/Fibrosis	[48]
miR-193b	↑ (pl)				[54]
miR-191	↑ (pl)				[54]
miR-140-3p	↑ (pl)				[54]
miR-454	↑ (pl)				[54]
miR-574	↑ (pl)				[54]
miR-885-5p	↑ (pl)				[54]
miR-886-3p	↑ (pl)				[54]
miR-27b	↓ (pl)				[54]
miR-23a/b	↓ (*m*h, h)	MEF2	↑ CELF1	Arrhythmias/Fibrosis	[48]
miR-208a	↑ (sk)		↓ several transcripts	Myofiber atrophy and hypertrophy	[55]
miR-381	↑ (sk)		↓ several transcripts	Myofiber atrophy and hypertrophy	[56]
miR-193b-3p	↓ (sk)		↑ several transcripts	Myofiber atrophy and hypertrophy	[55]
miR-7	↓ (*d*m, sk)				[52]
miR-10	↓ (*d*m, sk)				[52]
miR-15a	↓ (sk)		↑↓ several transcripts		[53]
miR-22	↓ (sk)		ERBB3		[53]
miR-155	↑ (sk)		↑↓ several transcripts		[53]
miR-222	↑ (sk)		↑↓ several transcripts		[53]
miR-381	↑ (sk)		↑↓ several transcripts		[53]
miR-411	↑ (sk)		↑↓ several transcripts		[53]
miR-200	↓ (bl)			Cancer	[57]
miR-241	↓ (bl)			Cancer	[57]
***DM2***					
miR-34a-5p; -34b-3p; -34c-5p	↑ (sk)		↓ several transcripts	Myofiber atrophy and hypertrophy	[55]
miR-146b-5p	↑ (sk)		↓ several transcripts	Myofiber atrophy and hypertrophy	[55]
miR-208a	↑ (sk)		↓ several transcripts	Myofiber atrophy and hypertrophy	[55]
miR-221-3p;	↑ (sk)		↓ several transcripts	Myofiber atrophy and hypertrophy	[55]
miR-381	↑ (sk)		↓ several transcripts	Myofiber atrophy and hypertrophy	[55]
miR-125-5p	↓ (sk)		↑ several transcripts	Myofiber atrophy and hypertrophy	[55]
miR-193a-3p; -193b-3p	↓ (sk)		↑ several transcripts	Myofiber atrophy and hypertrophy	[55]
miR-387a-3p	↓ (sk)		↑ several transcripts	Myofiber atrophy and hypertrophy	[55]
miR-1	↓ (*m*h, h)				[51]

DM1 and DM2 studies have been performed on several different disease models and samples: (sk) human skeletal muscle, (pl) human plasma, (se) human serum, (bl) human blood, (*m*h) mouse heart, (h) human heart, (*d*m) *Drosophila* muscle. miRNAs or transcripts/protein factors were found: (↑) upregulated, (↓) downregulated, (≠ cd) altered cellular distribution.

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
