# Peer review of "MicroRNA-Based Therapeutic Perspectives in Myotonic Dystrophy"

_ijms, 2019, doi:10.3390/ijms20225600_

Round 1

Reviewer 1 Report

This review by López Castel et al. entitled “microRNA-based therapeutic perspectives in Myotonic dystrophy” seeks to outline the current landscape of miRNA and their application to the neuromuscular disease, myotonic dystrophy. The authors outline the basic molecular and phenotypic aspects of the disease along with the roles of MBNL1 and CELF1 in the various DM cell and animal models. Through the use of detailed and well laid out figures along with descriptive text, the authors have outlined the basics of miRNA biology along with the current understanding of the role the miRNA plays in DM.  They have also provided an outline of the current status of miRNA-based research and the prospect for the use of the technology as a future therapeutic.

The manuscript covers the topic of miRNA and DM1 in a detailed manner and it is clear that the authors understand the field and have a detailed knowledge of the experiment and findings. They are well-known researchers in the DM field and are well suited as potential reviewers of the subject matter. The figures are well illustrated and convey the concepts without being overly technical or too cluttered or detailed. The manuscript provides a well-detailed basis for a review of miRNA-based therapeutics in myotonic dystrophy. However, the language and grammar hamper the ability of a reader unfamiliar with the source material to comprehend the subject matter.  The manuscript requires substantial revision with a focus on language and comprehension.

Some of the concerns and comments on the manuscript include

Language syntax, grammar, comprehension and flow issues. Specific examples include but are not limited to: Specify mouse or human cells when discussing a particular experiment Avoid long sentences with multiple concepts, split long sentences into shorter sentences (but not too short) that focus on one particular concept. Specify wildtype or DM mouse when using the term “mouse” to describe findings Authors need to introduce briefly the various mouse models before using them within a sentence. Examples include The HSALR mouse model could be described as a DM1 model but it is important for reader to understand the model lacks DMPK gene context and is restricted to muscle tissue. Readers need to know that the EpA960 mouse model is inducible and how it is induced before they can understand its use in miRNA analysis (Line 247-248).

Line 40 - more appropriate to say RNA rather than mRNA line 49, do the authors actually mean ‘post-transcriptional splicing’?

Lines 56-59, would be good to provide the specific splicing events associated with each symptom

Line 217 – the authors could contrast the findings from the miRNA-induced MBNL overexpression with those in which MBNL is overexpressed by other means. Is there a difference in the tolerability between miRNA overexpression versus other means that favors the miRNA approach?

It is difficult to parse many of the sentences in the review in order to understand the author’s underlying meaning – When it is possible to parse the sentences the meaning itself is correct but comprehension of the sentence requires knowledge of the field. Differentiate between experimental observation and the methods used to obtain those observation. When referencing multiple papers to describe a concept or findings, it is important to provide references within follow-up statements so readers don’t have to go back and hunt for the original reference.

Line 238 – missing reference for statement that begins “Similarly, ..” Sentences often missing noun or subject of sentence making it difficult to understand the intent or follow the flow of the author’s thoughts. The inclusion of sub-headings to help with flow of paragraphs could be beneficial.

Compare and contrast the miRNA-based approach with other therapeutic approaches. Always include description of the system under which observations and experiments are being conducted.

Line 254 – Is this result from the EpA960 mouse or another inducible system?

Line 269 – What type of DM1 cell lines? Include phenotypic outcomes or note their absence when describing miRNA experiments.

Line 249 – What is the overall effect of the increased CELF1 levels?

Line 252 – What are the recovered phenotypes How were ASB2 target levels and mIRA established as functionally relevant interactions for DM muscle pathology – was this finding based solely upon correlation with DM samples or upon correlation with muscle histopathology – important information for readers to know Avoid generic statements regarding findings that lack specificity

Line 27 - and in additional DM studies – heart or muscle specific, what type of model system, patient tissues  …? Avoid generic phrasing such as “in other muscle diseases.” The inclusion of specific examples will help the reader to place the supporting evidence in context of the DM results.

Line 309 Provide specific examples of contrasting studies The authors could provide further speculation on the differences in miRNA candidate biomarkers between DM1 and DM2 – line 339

Line 358 – comment on why DM studies lag behind other diseases.

Line 392 – Are these findings in the context of DM or other diseases? End longer paragraphs with a summary sentence that leaves the reader with the main take away point of the paragraph and leads the reader into the next section. Provide further details on the mechanism of azobenzene and other small molecules.

Line 464 – It would be interesting to provide a short paragraph outlining the findings from other repeat expansion disorders Concluding paragraph is missing that summarizes the results of the entire review as well as provide the outline the future directions of the field.

Minor and other corrections include line 70, ‘half-life’ should be ‘life-span’ line 99, ‘3 UTR’ should be ‘3’ UTR’ lines 144-145 and 167, Figure 1 legend ‘MiRNA’ should be replaced with ‘miRNA’ line 210, ‘HSALR’ HSA should be italicized: ‘HSALR’ line 227, beginning the sentence with ‘MiRNA’, it should be spelled out ‘Micro RNA’

Figures are good but could include additional information. Figure 1 missing labelling of ribosome in figure. Label basic elements of mRNA. Links 1,2,3 elements in figure with text in figure legend. Could be enhanced by specific examples in figure legend for misregulation or therapeutic strategies with references. Figure legend is a little too long consider shortening. Figure 2 would be enhanced with a short description of the effects of the miRNA findings. Split table into direct vs indirect and describe in figure legend (information is already there just needs to be edited to highlight difference and for comprehension). Include target for each miRNA and/or if they are focused on MBNL or CELF pathways.

Reviewer 2 Report

In this review, the authors have introduced potential therapeutic approaches using microRNAs for myotonic dystrophies. The reviewer has raised some points that need to be addressed.

Page 1, line 14-17, these sentences are obscure since it is unclear what MBNL and CELF1 are doing in DM1. Additionally, decrease of MBNL and increase of CELF1 do not suggest that DMs could be cured. These sentences should be revised and described in detail.

Page 1, line 30-32, the references are required.

Page 1, line 42-44. please explain in detail why MBNL and CELF1 are critical for DM.

Page 1, line 44-53, there is a lack of evidence for the reason to specifically focus on MBNL and CELF1 as the therapeutic targets for DM. More references and explanation would be required.

Page 2, line 61-62, the references are required.

Page 2, line 66, the description of "KO Mbnl1" is not general. It should be revised.

Page 2, line 68-71 and line 72-74, the references are required.

Page 2, line 83-85, why were these sentences highlighted?

Page 3, line 97, “mild” is a typo. Please revise it.

Page 3, line 111-114, this sentence is obscure to describe the reason why microRNAs are suitable as therapeutic target. Detailed explanation with references are required.

Page 5, line 176-182, reference number(s) should be inserted.

Page 6, line 215-216, please describe the design of experiments. Were MBNLs over-expressed by miRNA inhibition?

Page 6, line 232-240, the references are required.

Page 7, line 250-251, the references are required.

Page 8, line 313, “mi16” should be revised to “miR-16”.

Page 8, line 314, “preclding” is a typo. Please revise it.

Page 10, line 355, the description of “technical breakthroughs” was obscure. The authors need to describe the technical issues for miRNA-based therapies more specifically.

Page 11, line 368-369, the references are required.

Page 11, line 405-407, references and detailed explanation are required.

Page 12, line 424-426, references are required.

Reviewer 3 Report

Authors reviewed evidence of microRNA in the pathomechanism of myotonic dystrophy and its potential application in therapy. The review is well-organized throughout.

As the authors pointed out, while microRNA-based therapy is quite attractive, there are many tasks to be solved, especially for multi-organ disease such as DM. Thus, I think they may give a brief comment on future research perspectives at the end of the paper.

Minor point

Line 97: I think "mild 90s" is a misspell of "mid 90s".

Round 2

Reviewer 1 Report

The revised manuscript by López Castel et al entitled “microRNA-based therapeutic perspectives in Myotonic dystrophy” outlines the current miRNA field in relation to the neurodegenerative disease myotonic dystrophy (DM). The authors provide context for the miRNA discussion by introducing the molecular, cellular and disease phenotypes of DM along with the roles of two key pathogenic factors: MBNL1 and CELF1. The role of miRNA in the disease process as well as a potential therapeutic avenue are discussed in detail alongside detailed and well laid out figures.  They have provided a good overview of miRNA-based DM research and its potential use a therapeutic agent.

The revised manuscript maintains the excellent coverage of the topic of miRNA in relation to DM and has addressed the majority of the concerns regarding language, comprehension and flow. While some language issues still remain (see attached list), they do not interfere with the comprehension of the subject matter and are correctable. The authors have satisfactorily addressed the reviewers concerns and have produced a revised manuscript that will serve as an excellent review of the therapeutic potential of miRNA-based therapeutic for myotonic dystrophy.

Author Response

Dear Reviewer,

thanks for your careful review of language, comprehension and flow of the manuscript.

We have already included all your text suggestions into the new version of the manuscript